# Plug-and-Play Self-Supervised Denoising for Pulmonary Perfusion MRI

**DOI:** 10.3390/bioengineering12070724

**Published:** 2025-07-01

**Authors:** Changyu Sun, Yu Wang, Cody Thornburgh, Ai-Ling Lin, Kun Qing, John P. Mugler, Talissa A. Altes

**Affiliations:** 1Department of Chemical and Biomedical Engineering, University of Missouri, Columbia, MO 65211, USA; ywy6y@missouri.edu; 2Department of Radiology, University of Missouri, Columbia, MO 65211, USA; cthornburgh@health.missouri.edu (C.T.); ai-ling.lin@health.missouri.edu (A.-L.L.); altest@health.missouri.edu (T.A.A.); 3Division of Biological Sciences, University of Missouri, Columbia, MO 65211, USA; 4Institute for Data Science and Informatics, University of Missouri, Columbia, MO 65211, USA; 5Department of Radiation Oncology, City of Hope National Medical Center, Duarte, CA 91010, USA; kqing@coh.org; 6Department of Radiology and Medical Imaging, University of Virginia, Charlottesville, VA 22908, USA; jpm7r@virginia.edu

**Keywords:** pulmonary perfusion MRI, denoising, self-supervised learning, plug-and-play, deep learning

## Abstract

Pulmonary dynamic contrast-enhanced (DCE) MRI is clinically useful for assessing pulmonary perfusion, but its signal-to-noise ratio (SNR) is limited. A self-supervised learning network-based plug-and-play (PnP) denoising model was developed to improve the image quality of pulmonary perfusion MRI. A dataset of patients with suspected pulmonary diseases was used. Asymmetric pixel-shuffle downsampling blind-spot network (AP-BSN) training inputs were two-dimensional background-subtracted perfusion images without clean ground truth. The AP-BSN is incorporated into a PnP model (PnP-BSN) for balancing noise control and image fidelity. Model performance was evaluated by SNR, sharpness, and overall image quality from two radiologists. The fractal dimension and k-means segmentation of the pulmonary perfusion images were calculated for comparing denoising performance. The model was trained on 29 patients and tested on 8 patients. The performance of PnP-BSN was compared to denoising convolutional neural network (DnCNN) and a Gaussian filter. PnP-BSN showed the highest reader scores in terms of SNR, sharpness, and overall image quality as scored by two radiologists. The expert scoring results for DnCNN, Gaussian, and PnP-BSN were 2.25 ± 0.65, 2.44 ± 0.73, and 3.56 ± 0.73 for SNR; 2.62 ± 0.52, 2.62 ± 0.52, and 3.38 ± 0.64 for sharpness; and 2.16 ± 0.33, 2.34 ± 0.42, and 3.53 ± 0.51 for overall image quality (*p* < 0.05 for all). PnP-BSN outperformed DnCNN and a Gaussian filter for denoising pulmonary perfusion MRI, which led to improved quantitative fractal analysis.

## 1. Introduction

Pulmonary perfusion is a critical factor in oxygen delivery. Assessment of pulmonary perfusion is important in pulmonary arterial and parenchymal diseases. However, one of the challenges of pulmonary MRI is low proton density and short T2* of lung, which lead to a low signal-to-noise ratio (SNR) [1]. Pulmonary dynamic contrast-enhanced (DCE) MRI is clinically useful for assessing pulmonary perfusion, but challenged by its low SNR, breathing artifacts, and the need to process large temporal 3D datasets, especially for high spatial resolution imaging. Quantitative dynamic perfusion MRI requires temporal series [1,2]; thus, perfusion images at peak enhancement [3] remain clinically utilized for visual assessment of diseases. Recently, fractal analysis has been shown to be effective for lung structure analysis in chest CT [4] and MRI myocardial perfusion [5]. If pulmonary perfusion images with higher SNR can be obtained, fractal analysis and k-means clustering [6] could be applicable for analyzing perfusion images.

To analyze the perfusion images, denoising is an important pre-processing step [2]. Supervised deep learning (DL) networks have been shown to outperform conventional denoising methods, but typically require a training set of high quality images (“clean” images) [7]. Due to the dynamics of pulmonary perfusion, it is not possible to acquire high spatial and temporal resolution, high SNR MR images for training. Additionally, the assumption of pixel-wise independent noise is not met due to the spatially varying noise caused by intensity inhomogeneity, parallel imaging, or surface coil-based acquisition [8]. The noise in pulmonary perfusion MRI was particularly spatially dependent because of the strong difference between low-intensity lung tissues and high-intensity other tissues [9]. While the asymmetric pixel-shuffle downsampling (PD) blind-spot network (AP-BSN) [10,11] has been recently shown to effectively remove spatial-corrected noise using self-supervised learning (without paired clean images), the process may induce loss of image fidelity [11], which is important for medical imaging.

We sought to develop and evaluate a self-supervised deep learning framework for denoising pulmonary DCE MR images. Specifically, we integrate an AP-BSN within a plug-and-play (PnP) framework [12], incorporating the original noisy image to enhance image fidelity, which is a critical aspect for medical imaging applications, and without requiring clean reference data. The model was trained on prospectively acquired 3D lung MRI datasets using 2D slices as inputs and evaluated through blinded radiologist scoring and fractal analysis of perfusion complexity, which enables pixel-by-pixel estimation of perfusion image complexity while mitigating potential bias introduced by noise. In this technical proof-of-concept study, we show that PnP-BSN could improve image SNR without significant loss of sharpness and enhance pulmonary perfusion analysis.

## 2. Materials and Methods

This retrospective study was performed in accordance with an approved protocol by the University of Missouri institutional review board (IRB) for human research.

### 2.1. Study Sample and Imaging Protocol

Data from a single center was included in the study (University of Missouri, Columbia, MO, USA). Between August 2019 and January 2023, 37 patients underwent perfusion MRI of the chest with time-resolved angiography with interleaved stochastic trajectories (TWIST). Details of the patient demographic information are summarized in Appendix A. No data used in this study were previously reported.

In total, perfusion MRI scans were acquired from 37 patients, including 5 scanned using a 3.0 T scanner (MAGNETOM Vida, Siemens Healthcare, Erlangen, Germany) and 32 scanned using a 1.5 T scanner (MAGNETOM Aera, Siemens Healthcare, Erlangen, Germany). For each individual, pulmonary perfusion images were acquired with the TWIST sequence. A gadolinium-based contrast agent (0.1 mmol/kg) was administered intravenously, and image acquisition was initiated shortly after injection, timed to capture the peak pulmonary perfusion phase during breathholds. The 37 subjects were randomly divided 80:20 into training and independent testing datasets, with 29 subjects (3448 2D slices) for training and 8 subjects (1124 2D slices) for testing. For the training set, TR was 2.31–2.93 ms, TE 0.87–1.01 ms, flip angle 18.2–30.2°, spatial resolution 1.09–1.41 mm, slice thickness 0.63–2.85 mm, bandwidth 440–994 Hz/pixel, and 81–153 slices. For the testing set, TR was 2.30–2.76 ms, TE 0.88–0.94 ms, flip angle 19.7–32.7°, spatial resolution 1.14–1.40 mm, slice thickness 0.62–2.92 mm, bandwidth 424–992 Hz/pixel, and 83–157 slices. Maximum intensity projection (MIP) images were calculated during post-processing. More details of time-resolved imaging with stochastic trajectories acquisition (TWIST) sequence parameters are provided in Appendix A.

### 2.2. Model Design

The overall pipeline is shown in Figure 1A. PnP-BSN is developed using AP-BSN in a PnP model using alternating direction method of multipliers (ADMM). Each two-dimensional (2D) coronal slice is denoised using the PnP-BSN model. The MIP image is derived from the denoised images. Fractal analysis is employed to calculate the fractal dimension of the pulmonary perfusion images for quantifying the complexity of the pulmonary perfusion image. Given that fractal analysis is sensitive to noise, it also serves to illustrate the effectiveness of denoising methods for improving the quantitative analysis. K-means clustering was applied to the denoised images to segment the lung region into three classes.

BSN is one of the representative self-supervised networks to reconstruct a clean pixel from the surrounding noisy pixels without referring to the exact input pixel (Figure 1B). However, due to the spatially correlated MRI noise, the performance of BSN alone will be suboptimal. We sought that by combining PD into BSN, which break down the spatial correlation of the noise, BSN can be generalized to actual acquired MRI data. By adopting a larger PD stride in training and smaller stride in testing, the asymmetric strategies will reach a good balance of reconstructing details [11]. More network structure and training hyperparameters of AP-BSN (Figure 1C) for pulmonary perfusion MRI are shown in Appendix A.

The proposed PnP-BSN model (Figure 1A) is expressed in Equation (1) as:(1)$$arg\underset{x,z}{\min}\frac{1}{2}||y-x{\left|\right|}^{2}+\frac{\rho }{2}||x-z{\left|\right|}^{2},\;\mathrm{subject}\;\mathrm{to}\;z=D_{\sigma }\left(x+u\right),$$
where y is the original noisy image, x is the denoised image, ρ is the weighting parameter to balance the data fidelity and the image prior to using AP-BSN, $D_{\sigma }$ in (1) is the pretrained AP-BSN denoiser using noisy pulmonary perfusion MR images, z is an auxiliary variable, and u is the dual variable. The first term enforces the denoised image x to be close to the noisy image y, the second term is a regularization term that enforces the output of the denoising network $D_{\sigma }$ to be close to the auxiliary variable z. x is solved using ADMM. ρ is selected as 1 to balance the noise control and blurring effects as $x^{\left(k+1\right)}=\frac{y+\rho \left(z^{\left(k\right)}-u^{\left(k\right)}\right)}{1+\rho }$, where k represents the iterative index ranging from 1 to 3. Here, the PnP-BSN method integrates AP-BSN as its denoiser. (in the second term of Equation (1)) and enforces image fidelity using the original noisy image (in the first term of the iterative PnP algorithm). The detailed algorithm of PnP-BSN is provided in Appendix A.

### 2.3. Perfusion Complexity Analysis Using Fractal Analysis

We explored the use of fractal dimension (FD) derived from pulmonary perfusion image textures to quantify image complexity. FD maps were generated through fractal analysis [4,5,13], where the FD values illustrate texture complexity; a lower FD indicates reduced complexity, with 2.0 as the baseline [5,14]. Detailed methodology of the fractal analysis is described in Appendix A.

### 2.4. Data Processing and Evaluation

TWIST images were background subtracted. Peak perfusion images were utilized for analysis. The noisy images were denoised by different denoising methods. The input data for the network was one 2D noisy perfusion image after background subtraction. After AP-BSN was trained, the network was used as the image prior in Equation (1). AP-BSN training and testing were performed with PyTorch on a 48G GPU server (RTX A6000; NVIDIA). The model was trained for 20 epochs (approximately 23 h), and the inference time of AP-BSN per image was ~8 s. For comparing the denoising performance, DnCNN was applied using the pretrained model provided in MATLAB R2023a (Mathworks Inc., Natick, MA, USA), and a Gaussian filter with optimal performance was used after testing and iterating different noise variations to find the optimal balance in denoising and blurring. Two blinded radiologists scored the denoised images on a 1–5 scale (1 is the worst and 5 is the best), assessing SNR, sharpness, and overall image quality. Detailed radiologist scoring guidelines are provided in Appendix A. Inter-observer agreement was analyzed by calculating the correlation, and method differences were tested with the Wilcoxon signed rank test (*p* < 0.05 was considered statistically significant).

## 3. Results

### 3.1. Denoising Performance and Fractal Analysis

Figure 2A shows an example of a patient’s MIP images that were denoised using DnCNN, a Gaussian filter, and PnP-BSN. PnP-BSN shows better noise control and lower sharpness loss in the images. The FD map is calculated using the MIP images and shown in Figure 2B, which further shows the improvement in the quantitative fractal analysis using PnP-BSN, as the vessel edges and complexity are better illustrated. Figure 2C shows overlaid MIP images with the fractal analysis for better visualization of lung structure and edge clarity.

### 3.2. Patient Example: Pulmonary Embolism Case Study

Figure 3 presents denoised images, k-means segmentation results, and MIP images for a 54-year-old male patient with pulmonary embolism. In Figure 3A, the original noisy image and denoised images produced by DnCNN, Gaussian filter, and PnP-BSN are shown. PnP-BSN demonstrated superior visualization of the pulmonary structure with balanced noise control and detail preservation compared to DnCNN and Gaussian filter. K-means analysis (Figure 3B) segmented the images into three classes (low-, medium-, and high-intensity regions) based on pixel intensity, with PnP-BSN providing improved delineation of high-intensity regions. Figure 3C displays MIP images generated from 20 slices, where PnP-BSN denoising yielded enhancing image clarity and recovering vessel structures obscured by noise. All slices are shown in Appendix A. An example from a pediatric patient is presented in Appendix A, illustrating the improved vascular structures by PnP-BSN compared to other denoising methods.

### 3.3. Comparison with Reference Clinical Imaging

In Figure 4, we compare the noisy and PnP-BSN-denoised MIP perfusion image using the T2-weighted MR image and chest X-ray image at the same location. The FD maps calculated from noisy and PnP-BSN-denoised images are shown, respectively, which reveals the decay in perfusion complexity in the outer region of the left lung by pneumonia. The FD map generated by the PnP-BSN-denoised MIP image better illustrates the low complexity of the left lung outer region, as indicated by the black arrows. Without denoising, the noisy image shows high perfusion complexity of the lungs because the noise induced errors. With PnP-BSN, the perfusion complexity mapped from the denoised image shows the disease clearly and quantifies the complexity of the pulmonary perfusion image patterns.

### 3.4. Evaluation of Image Quality

Figure 5 shows score bar plots of SNR, sharpness, overall image quality for DnCNN, Gaussian, and PnP-BSN-denoised images. Eight out of the thirty-seven individuals in the independent test set with images from the peak perfusion measurement were evaluated by two radiologists. The expert scoring results for DnCNN, Gaussian, and PnP-BSN were 2.25 ± 0.65, 2.44 ± 0.73, and 3.56 ± 0.73 (*p* < 0.05) for SNR; 2.62 ± 0.52, 2.62 ± 0.52, and 3.38 ± 0.64 (*p* < 0.05) for sharpness; and 2.16 ± 0.33, 2.34 ± 0.42, and 3.53 ± 0.51 (*p* < 0.05) for overall image quality. There was good agreement between observer ratings for the three metrics of SNR (r = 0.72), sharpness (r = 0.67), and overall image quality (r = 0.77).

## 4. Discussion and Conclusions

While pulmonary contrast perfusion MRI using the TWIST sequence provides perfusion assessment of the lungs, it suffers from a low SNR due to low proton density and short T2* lung parenchyma and image quality degradation from motion [9]. The low SNR of the pulmonary perfusion MRI affects the qualitative and quantitative analysis of perfusion MRI [15]. A denoising process is commonly used for pre-processing lung perfusion MRI [2], but conventional denoising algorithms often introduce blurring and do not fully suppress noise. Many medical imaging deep learning techniques utilize supervised learning which require training data sets that are labeled or acquired with the desired image quality. The fundamental issues with pulmonary contrast perfusion MRI are the rapid circulation of contrast and the necessity for high spatiotemporal resolution imaging, which means it is not possible to obtain images of the desired image quality (clean images) to use as training data for deep learning algorithms.

In this proof-of-concept study, we developed a PnP AP-BSN self-supervised learning algorithm without paired clean images [11,16,17] for pulmonary contrast perfusion denoising. We incorporated a PnP model [12] to balance the noise control and image fidelity, which improves the fidelity of the denoised images. Despite the relatively small training set (29 exams), our self-supervised learned AP-BSN [11] plugged into the PnP model was effective for denoising pulmonary perfusion MRI and maintaining image fidelity based on the scores from two blinded radiologists. The image quality improvement from our self-supervised learning denoising method led to improved quantitative analyses using fractal analysis [4,14] and k-means clustering.

Prior studies used a Gaussian denoising algorithm for pulmonary contrast perfusion MRI [2]. The Gaussian filter can induce blurring artifacts on pulmonary MR images. The varied noise distributions on pulmonary perfusion MR images may violate the noise assumptions of deep learning-based Gaussian denoising methods, such as DnCNN. Our method showed greater image sharpness and better noise control as compared with DnCNN. Our algorithm appears to be robust to different field strengths (1.5T and 3T), although this would need to be verified in a larger study. Our method also could be generalized to other MRI data where the acquisition of clean training data is difficult or impossible, such as in myocardial perfusion MRI. Fractal analysis has been applied to chest CT [4] for assessing lung structures, but it has not been applied to lung perfusion MRI for perfusion complexity assessment. Our proof-of-concept study of fractal analysis in lung perfusion MRI shows its potential use for assessment of pulmonary perfusion complexity.

A limitation of the current study is the small number of examinations in the training (*n* = 29) and testing datasets (*n* = 8). However, it is promising that the self-supervised technique achieved good performance despite a small training dataset that contained a variety of ages and diseases. Current training input is a 2D perfusion MR image after background subtraction due to the GPU processing requirements of 3D datasets. Further training with a 3D network may improve the denoising performance of PnP-BSN. While the reader scores in this study were based on side-by-side comparisons with the original noisy images to mitigate potential bias, the possibility remains that the deep learning model could affect subtle diagnostic features. Further research using larger training and testing datasets that include confirmed pathology cases or lung phantom study with simulated perfusion patterns and realistic noise models are needed to more rigorously evaluate the diagnostic performance and generalizability of the proposed method.

Future work could combine the proposed denoising and fractal analysis methods with temporal acquisitions and other quantitative measurements [2] such as first moment transit time, time to peak, pulmonary blood flow, etc. The model can also be applied to pulmonary perfusion MRI for validating lung diseases such as idiopathic pulmonary fibrosis, pulmonary vascular diseases, cystic fibrosis, and pulmonary arterial hypertension. Future work should extend the processing to other MR images with low SNR. While our study focuses on pulmonary perfusion, the use of fractal analysis to quantify image complexity may be applicable to other tissues, such as the brain [18,19,20]. Conventional methods such as BM3D remain widely used in medical image denoising [21] due to their robustness and interpretability, while unsupervised denoising networks like the deep image prior [22] have demonstrated the potential for leveraging only the noisy image itself for denoising. More recently, a range of deep learning-based methods have been proposed for denoising in medical imaging [23]; however, relatively few studies have focused specifically on pulmonary MRI when paired clean images are unavailable, highlighting the need for a self-supervised learning approach. Our work demonstrates the feasibility of self-supervised denoising for pulmonary perfusion MRI. Unlike methods such as Gaussian filtering, BM3D, or supervised deep learning approaches that require paired high-quality training data, and unlike the blind-spot network [16] that assume pixel-independent noise, our approach can denoise without clean reference images and adapt to the spatially varying noise characteristics of pulmonary imaging. We address the challenge from spatially varying noise by employing AP-BSN within a PnP framework to improve image fidelity. The subsequent fractal analysis of the denoised images provides a quantitative assessment, revealing the fractal dimension in pulmonary perfusion MRI and suggesting that denoising enhances quantitative pulmonary image analysis.

In conclusion, to overcome the inherent inability to acquire high SNR training pulmonary contrast perfusion MR images, we developed a self-supervised learning framework to denoise pulmonary contrast perfusion MR images and found it improved SNR and maintained image sharpness better than Gaussian denoising algorithms. Our self-supervised learning framework and the post-denoising perfusion analysis (fractal analysis and k-means clustering) are promising methods for improving pulmonary perfusion MRI image quality and quantification. Additionally, this self-supervised learning framework could be applied to other situations in which it is not feasible to acquire high-quality training data.

## Figures and Tables

**Figure 1 bioengineering-12-00724-f001:**
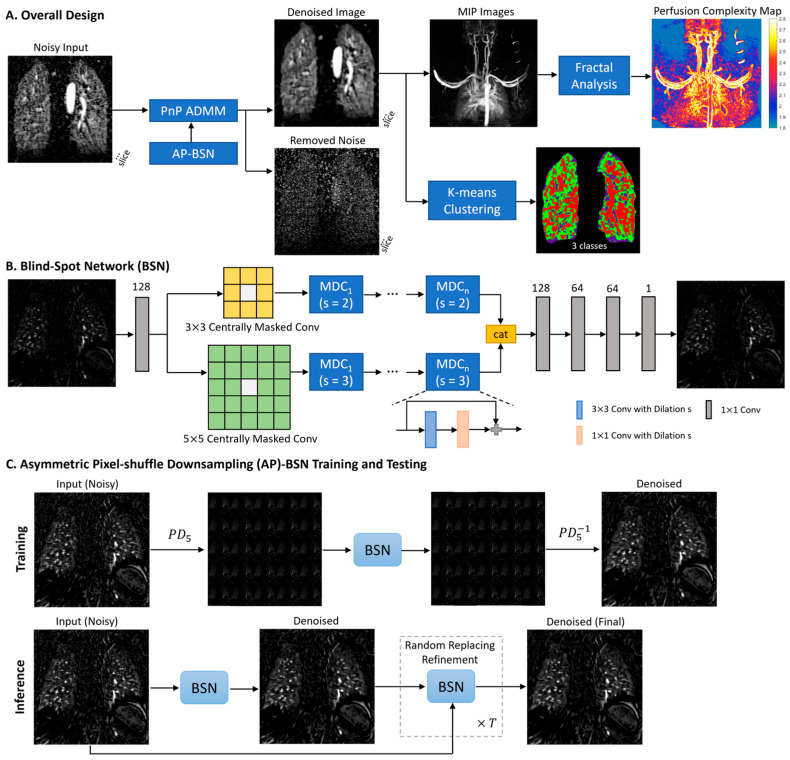
(**A**) Illustration of the processing pipeline of the plug-and-play blind-spot network (PnP-BSN) for denoising lung DCE MR images, followed by fractal analysis of the maximum intensity projection (MIP) to quantify the complexity of the perfusion image. Each 2D noisy slice serves as an input to the PnP-BSN model (Equation (1)), which utilizes asymmetric pixel-shuffle downsampling (PD) BSN within the PnP framework. (**B**) The structure of the AP-BSN features two distinct convolution branches, followed by centrally masked convolution blocks (3 × 3 and 5 × 5), dilated convolution blocks with skip connections (s = 2 and s = 3), concatenation of two branches (cat) and 1 × 1 convolution blocks. (**C**) Illustration of the AP-BSN structure, which utilizes PD = 5 to replicate the images, followed by BSN and inverse PD during training. During inference, an iterative random replacement refinement is applied T = 16 times. More details of AP-BSN can be found in Appendix A. PnP = plug-and-play, BSN = blind-spot network, MIP = maximum intensity projection, 2D = two-dimensional, AP = asymmetric pixel-shuffle downsampling.

**Figure 2 bioengineering-12-00724-f002:**
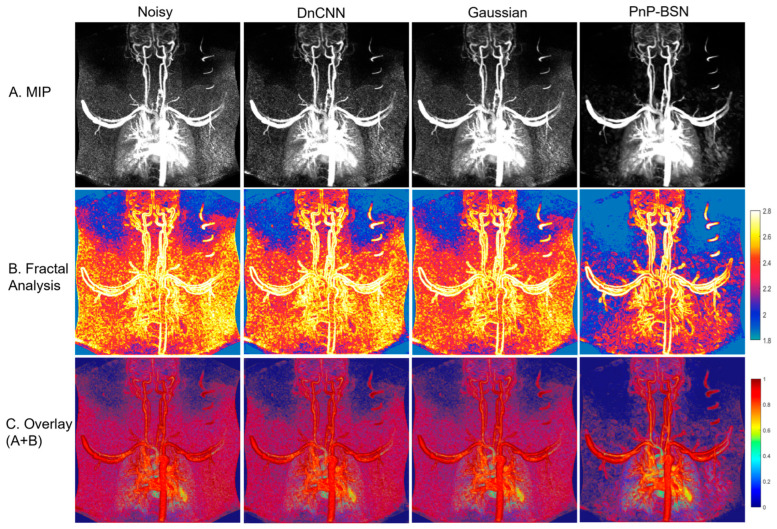
(**A**) Example MIP images derived from the 3D noisy pulmonary perfusion MRI, Gaussian filtered images, DnCNN-denoised images, and PnP-BSN-denoised images. (**B**) Fractal dimension of the MIP images demonstrates the complexity of pulmonary perfusion images for evaluating and demonstrating the denoising performance of the different methods. The results indicate that PnP-BSN-denoised images improve edge detection and complexity quantification in pulmonary perfusion imaging. (**C**) Combined visualization of the original MIP images with fractal analysis offers a detailed perspective on lung structure and edge clarity.

**Figure 3 bioengineering-12-00724-f003:**
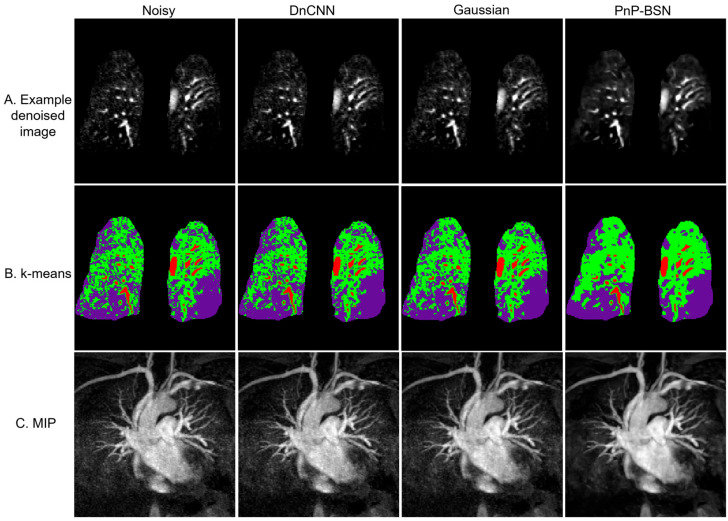
Example denoised images, k-means segmentation analysis, and MIP for a 54-year-old male patient with pulmonary embolism, imaged at 1.2 mm isotropic resolution. (**A**) Presentation of the original noisy image alongside denoised images produced by three methods (DnCNN, Gaussian, and PnP-BSN), with PnP-BSN demonstrating superior effectiveness in enhancing image clarity and detail. (**B**) The results of k-means analysis applied to the images in (**A**), segmenting the images into three classes (low-, medium- and high-intensity regions) based on pixel values. Red, green, and purple correspond to high-, medium-, and low-intensity regions, respectively. (**C**) MIP images generated from 20 slices (24 mm), providing a detailed view of the pulmonary structure.

**Figure 4 bioengineering-12-00724-f004:**
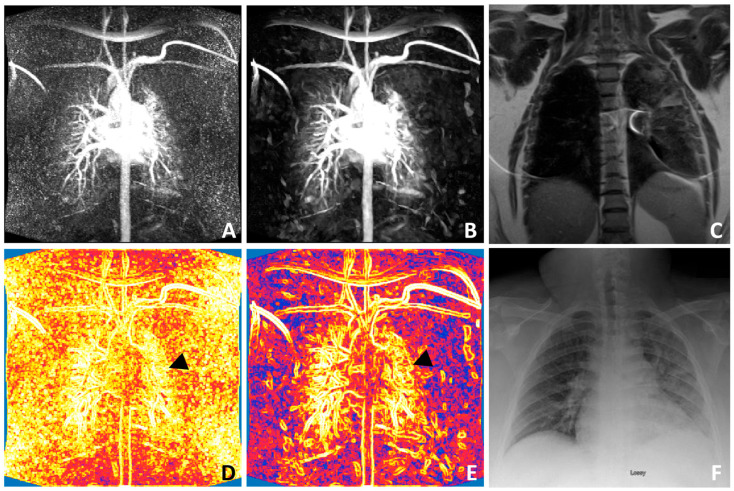
Example MIP images from a patient with pneumonia, using the T2-weighted MR image (**C**) and chest X-ray (**F**) image as references. The original noisy MIP image (**A**) and the PnP-BSN-denoised image (**B**) are shown, highlighting significant noise control by PnP-BSN. Perfusion complexity maps for the noisy (**A**) and denoised (**B**) images are presented in (**D**,**E**), respectively, demonstrating a decay in perfusion complexity in the outer region of the left lung that is better shown by the PnP-BSN-denoised image and validated by the reference images (**C**,**F**). PnP-BSN = plug-and-play blind-spot network.

**Figure 5 bioengineering-12-00724-f005:**
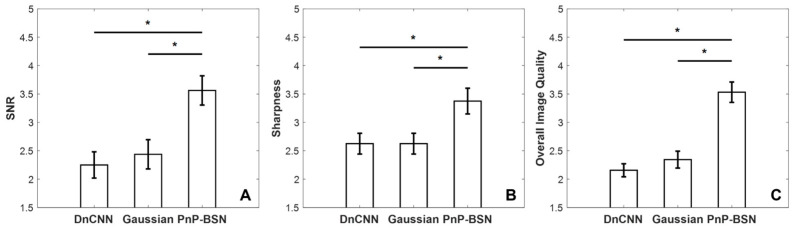
Quantitative comparisons of PnP-BSN, DnCNN, and Gaussian methods in terms of (**A**) signal-to-noise ratio (SNR), (**B**) sharpness, and (**C**) overall image quality for pulmonary perfusion images scored by two radiologists, demonstrating that PnP-BSN significantly outperformed the other methods across all metrics, with superior SNR (3.56 ± 0.73 for PnP-BSN vs. 2.44 ± 0.73 for Gaussian and 2.25 ± 0.65 for DnCNN, *p* < 0.05), sharpness (3.38 ± 0.64 for PnP-BSN vs. 2.62 ± 0.52 for Gaussian and DnCNN, *p* < 0.05), and overall image quality (3.53 ± 0.51 for PnP-BSN vs. 2.34 ± 0.42 for Gaussian and 2.16 ± 0.33 for DnCNN, *p* < 0.05). * indicates *p* < 0.05.

## Data Availability

The raw data presented in this article are not readily available due to patient privacy. Requests to access the code and dataset for research purpose should be directed to the corresponding author at csyfc@missouri.edu.

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
