# Peer review of "Plug-and-Play Self-Supervised Denoising for Pulmonary Perfusion MRI"

_bioengineering, 2025, doi:10.3390/bioengineering12070724_

Round 1
Reviewer 1 Report
Comments and Suggestions for Authors
The topic of the study is interesting and reads mostly fluently. However, there are certain parts that are misleading and not meaningful in terms of scientific analysis. The dataset is very small, which is one of the big concerns. Please see the comments.
- The size of the dataset raises concerns for the generalization of the study results.
- The format of the results section needs to be corrected.
- The complexity of the model will be negatively affecting the performances of the radiologists. They may not be detecting the small details on the images.
- Please clarify the following "low proton density and short T2* of lung".
- How "analysis of large temporal three-dimensional (3D) datasets " is a limitation?
- Please review the manuscript for language related errors.
- Please support your claim "typically require a training set of high quality images".
- Why denosing is innovative "An innovative feature of our approach is incorporating the original noisy image to enhance image fidelity, a critical aspect for medical imaging applications.". Quick literature search provided plenty of references 4+ years old with PnP approach.
- Regarding following statetement, you mentioned noise is not be exclusively studied for individual pixels? Please clarify. "which provides pixel-by-pixel perfusion image complexity under control of potential bias from noise "
- Please clarify what you mean "5 datasets were acquired using 3.0 T scanners ..." and "32 were acquired using 1.5 T scanners ". Do you mean patients were scanned with different scanner with associated number?
- You may not say "patient datasets" which is misleading.
- You cannot express the acquisition parameters with deviation which is misleading. Did you use different acquisition parameters for each patients or scanners?
- You cannot compare MRI acquisition parameters for significance! You acquire physical images with the parameters, and they can affect the images heavily with small changes. Please revise after discussing with the radiologists.
- Please provide more details regarding the training and performance evaluation of the DL models.
- Did you use quantitative analysis metrics? The sharpness, contrast improvement?
Author Response
We thank the reviewer for the time, effort, and attention given to our manuscript. The comments and suggestions were thoughtful and are addressed point-by-point below. We feel that addressing the reviewer’s concerns has significantly strengthened and improved the manuscript.
Reviewer #1 (Comments for the Author):
The topic of the study is interesting and reads mostly fluently. However, there are certain parts that are misleading and not meaningful in terms of scientific analysis. The dataset is very small, which is one of the big concerns. Please see the comments.
R1C1. The size of the dataset raises concerns for the generalization of the study results.
Response: Thank you for this comment. We appreciate the reviewer’s feedback and recognize that the relatively small dataset size being a limitation of this study. To more explicitly indicate this constraint, we have revised the text in the manuscript related to dataset size to be: “A limitation of the current study is the small number of exams in the training (n=29) and testing datasets (n=8),” which reflects our intention to clearly and rigorously disclose this limitation.
We would like to clarify that, although the number of unique subjects is limited, each dataset comprises full 3D lung image volumes. From the training datasets, a total of 3448 2D slices were extracted, and from the testing datasets, 1124 2D slices were included. While the number of patients remains limited, the large number of spatially diverse slices provides substantial variability, which helps improve the robustness and generalizability of the deep learning model.
We added this information in Methods Section as: “The 37 patient datasets were randomly divided 80:20 into training and independent testing datasets, with 29 subjects (3448 2D slices) for training and 8 subjects (1124 2D slices) for testing.”
R1C2. The format of the results section needs to be corrected.
Response: We appreciate the reviewer’s feedback regarding the format of the Results section. We have carefully reviewed this section for clarity, structure, and adherence to formatting guidelines. While the original content followed the journal’s requirements to the best of our understanding, we have made minor revisions to improve flow and consistency by adding subtitles. The added subtitles are as follows:
“3.1. Denoising performance and fractal analysis
3.2. Patient example: pulmonary embolism case study
3.3. Comparison with reference clinical imaging
3.4. Evaluation of image quality”
R1C3. The complexity of the model will be negatively affecting the performances of the radiologists. They may not be detecting the small details on the images.
Response: We appreciate the reviewer’s thoughtful comment regarding the potential impact of model complexity on radiologist performance.
1. The comparisons in our study were made against the original noisy images, allowing radiologists to evaluate whether any subtle anatomical or pathological features were lost after denoising. Any missed details in the denoised images were considered in the assessment and are reflected in the results.
2. As ground truth is inherently unavailable for contrast-enhanced perfusion MRI, particularly under clinical acquisition conditions, our method was designed using a self-supervised learning framework. This allows the model to learn denoising features directly from the data without relying on externally generated reference images.
We agree with the reviewer that additional clinical validation, particularly in patients with confirmed pathology, will be important in future work. Such studies will allow a more comprehensive understanding of how model complexity interacts with diagnostic performance and radiologist interpretation. We have added an explanation to this effect in the revised Discussion section.
“While the reader scores in this study were based on side-by-side comparisons with the original noisy images to mitigate potential bias, the possibility remains that the deep learning model could affect subtle diagnostic features. Further research using larger training and testing datasets that include confirmed pathology cases or lung phantom study with simulated perfusion patterns and realistic noise models are needed to more rigorously evaluate the diagnostic performance and generalizability of the proposed method.”
R1C4. Please clarify the following "low proton density and short T2* of lung".
Response: We thank the reviewer for requesting clarification. The phrase “low proton density and short T2* of the lung” refers to two key physical limitations in lung MRI. First, the lung parenchyma contains mostly air, resulting in low proton density and consequently low intrinsic MRI signal. Second, the numerous air-tissue interfaces in the lung cause strong local magnetic susceptibility gradients, which lead to rapid signal decay (short T2*), making it challenging to acquire high-quality images. These properties make lung MRI particularly difficult compared to other soft tissues.
R1C5. How "analysis of large temporal three-dimensional (3D) datasets " is a limitation?
Response: We thank the reviewer for pointing this out. To clarify, we did not intend to suggest that the analysis of large temporal 3D datasets is a specific limitation of our method. Rather, this was meant to highlight a general challenge associated with pulmonary DCE-MRI, particularly when aiming for high spatial and temporal resolution. The complexity and size of 4D datasets (3D volumes over time) present a practical obstacle for both manual and automated image processing workflows in clinical settings.
To clarify this point, we have revised this sentence in the Introduction section to be: “Pulmonary dynamic contrast-enhanced (DCE) MRI is clinically useful for assessing pulmonary perfusion, but it is challenged by low SNR, breathing artifacts and the need to process large temporal 3D datasets, especially for high spatial resolution imaging.
R1C6. Please review the manuscript for language related errors.
Response: We thank the reviewer for this helpful suggestion. We have carefully reviewed the manuscript for language-related errors and have revised the text to improve grammar. The revised version has been proofread to ensure that the language meets the standards of scientific writing. We hope these changes enhance the overall quality and presentation of the manuscript.
R1C7. Please support your claim "typically require a training set of high quality images".
Response: We thank the reviewer for this important point. We have clarified the statement in the manuscript and now support it with a citation. For example, in the widely cited work by Zhang et al. (DnCNN, IEEE TIP 2017), the authors trained their supervised denoising network using 384000 pairs of clean and noised images, highlighting the requirement for a large dataset of high-quality reference images. We have now included this reference in the revised manuscript to support the statement. We have added the reference to “typically require a training set of high quality images”.
R1C8. Why denosing is innovative "An innovative feature of our approach is incorporating the original noisy image to enhance image fidelity, a critical aspect for medical imaging applications.". Quick literature search provided plenty of references 4+ years old with PnP approach.
Response: We appreciate the reviewer’s comment and agree that the Plug-and-Play (PnP) framework is not a novel concept. We did not intend to present PnP as our innovation. Rather, our contribution lies in adapting the PnP framework within a self-supervised denoising pipeline using an asymmetric pixel-shuffle downsampling blind-spot network (AP-BSN), which we refer to as PnP-BSN for pulmonary MRI images. This represents a focused self-supervised learning method on dynamic contrast-enhanced pulmonary MRI, where ground truth is unavailable.
We have revised the manuscript to more accurately reflect that PnP is not novel and that our contribution is the specific integration of PnP with AP-BSN for this application. We also emphasize our combination of AP-BSN with PnP without requiring clean images as reference as follows:
“Specifically, we integrate an AP-BSN within a Plug-and-Play (PnP) framework [12] in-corporating the original noisy image to enhance image fidelity which a critical aspect for medical imaging applications and without requiring clean reference data.”
R1C9. Regarding following statement, you mentioned noise is not be exclusively studied for individual pixels? Please clarify. "which provides pixel-by-pixel perfusion image complexity under control of potential bias from noise"
Response: We thank the reviewer for the opportunity to clarify this sentence. The intent was not to suggest that noise was studied at the individual pixel level, but rather that noise is a confounding factor when computing pixel-level complexity using fractal analysis. Because fractal analysis quantifies local spatial complexity, it is sensitive to random fluctuations caused by image noise, which can artificially elevate the complexity metric. However, the noise in pulmonary perfusion MRI may not be pixel-wise independent noise.
By applying denoising prior to fractal analysis, we aim to reduce the bias introduced by noise, thus improving the accuracy of pixel-by-pixel assessment of perfusion heterogeneity. In the revised manuscript, we have reworded this sentence for clarity as follows:
“... which enables pixel-by-pixel estimation of perfusion image complexity while mitigating potential bias introduced by noise.”
R1C10. Please clarify what you mean "5 datasets were acquired using 3.0 T scanners ..." and "32 were acquired using 1.5 T scanners ". Do you mean patients were scanned with different scanner with associated number?
Response: We thank the reviewer for pointing this out. Yes, each dataset corresponds to a separate MRI scan from a unique individual. Specifically, 5 patients were scanned using a 3.0 T MRI scanner and 32 patients were scanned using a 1.5 T MRI scanner. We have revised the sentence in the manuscript to clarify this.
“In total, perfusion MRI scans were acquired from 37 patients, including 5 scanned using a 3.0 T scanner (MAGNETOM Vida, Siemens Healthcare, Erlangen, Germany) and 32 scanned using a 1.5 T scanner (MAGNETOM Aera, Siemens Healthcare, Erlangen, Germany).”
R1C11. You may not say "patient datasets" which is misleading.
Response: We thank the reviewer for this helpful clarification. To avoid this confusion, we have revised the wording throughout the manuscript to refer to “image acquisitions” or “scans” instead. Specifically, the revised sentence now reads:
“In total, perfusion MRI scans were acquired from 37 patients, including 5 scanned using a 3.0 T scanner (MAGNETOM Vida, Siemens Healthcare, Erlangen, Germany) and 32 scanned using a 1.5 T scanner (MAGNETOM Aera, Siemens Healthcare, Erlangen, Germany).”
R1C12. You cannot express the acquisition parameters with deviation which is misleading. Did you use different acquisition parameters for each patients or scanners?
Response: We thank the reviewer for this observation. Slightly different acquisition parameters were used across scans due to differences in protocol adjustments at the time of clinical imaging. We have revised the manuscript to report representative parameter ranges instead of mean ± SD: “For the training set, TR was 2.31-2.93 ms, TE 0.87-1.01 ms, flip angle 18.2-30.2 °, spatial resolution 1.09-1.41 mm, slice thickness 0.63-2.85 mm, bandwidth 440-994 Hz/pixel, and 81-153 slices. For the testing set, TR was 2.30-2.76cms, TE 0.88-0.94 ms, flip angle 19.7-32.7 °, spatial resolution 1.14-1.40 mm, slice thickness 0.62-2.92 mm, bandwidth 424-992 Hz/pixel, and 83-157 slices.”
R1C13. You cannot compare MRI acquisition parameters for significance! You acquire physical images with the parameters, and they can affect the images heavily with small changes. Please revise after discussing with the radiologists.
Response: We have removed the statistical comparison of acquisition parameters from the Supplementary Information Table S2.
R1C14. Please provide more details regarding the training and performance evaluation of the DL models.
Response: We thank the reviewer for this suggestion. In the revised manuscript, we have provided clear and explicit definitions for all scoring factors used in the radiologist-based evaluation. These details have been provided as Supplementary Information Text 4.
Three image quality criteria were assessed for each denoising method: signal-to-noise ratio (SNR), sharpness, and overall image quality. These were evaluated subjectively by two radiologists using a standardized 5-point Likert scale.
• SNR was defined as the radiologist’s perception of noise in the image, where a higher score indicates lower visible noise.
• Sharpness was defined as the perceived clarity of anatomical structures and the degree to which structural blurring was suppressed by the denoising algorithm.
• Overall image quality reflected a combined impression of noise, structural fidelity, and sharpness, using the original noisy image as a reference.
Scoring was performed using the following 5-point scale:
(1) Poor: high noise and/or significant blurring;
(2) Fair: reduced noise and/or significant blurring;
(3) Good: low noise but moderate blurring;
(4) Very good: low noise and well-controlled blurring;
(5) Excellent: no significant noise or blurring.
Radiologists viewed one PowerPoint slide per patient, which displayed all the slices of denoised images from the same subject but from four different methods (Methods 1–4) presented in random order from left to right. The original noisy image was also shown as a reference. Each method was scored independently in all three categories, and the scores were recorded using a structured Excel sheet to support later statistical analysis.
R1C15. Did you use quantitative analysis metrics? The sharpness, contrast improvement?
Response: We thank the reviewer for this insightful question. In this study, we did not use quantitative image quality metrics such as sharpness or contrast improvement, as no ground truth or clean reference images were available for pulmonary perfusion datasets. Given the nature of self-supervised denoising and the lack of standardized references in dynamic contrast-enhanced lung imaging, we instead relied on blinded radiologist scoring to evaluate image quality and diagnostic acceptability. However, as indicated in R1C14, the noisy images are always provided to the radiologists as reference for detecting the difference of the image details.
We agree with the reviewer that future work may benefit from incorporating additional quantitative metrics. However, designing such metrics is non-trivial in the absence of ground truth, especially since synthetic noise in phantom studies may not accurately reflect the complex noise patterns present in clinical acquisitions. Nonetheless, a well-controlled lung phantom study with simulated perfusion patterns and realistic noise models could offer a valuable complement to clinical evaluation, and we have noted this as a direction for future work in the revised Discussion section as: “Further research using larger training and testing datasets that include confirmed pathology cases or lung phantom study with simulated perfusion patterns and realistic noise models are needed to more rigorously evaluate the diagnostic performance and generalizability of the proposed method.”

Reviewer 2 Report
Comments and Suggestions for Authors
This study proposes a self-supervised denoising model based on a Plug-and-Play (PnP) framework incorporating the Asymmetric Pixel-shuffle Downsampling Blind-Spot Network (AP-BSN), aiming to enhance the image quality of pulmonary perfusion MRI. The manuscript effectively addresses the challenges of low signal-to-noise ratio (SNR) in dynamic contrast-enhanced (DCE) pulmonary MRI due to low proton density and short T2*, as well as the fundamental limitation of lacking “clean” images for supervised training. The model performs denoising without the need for clean image references by balancing noise control and image fidelity. Through radiologist-based scoring of SNR, sharpness, and overall image quality, as well as quantitative assessments using fractal dimension analysis and k-means clustering, the PnP-BSN approach demonstrates superior performance in both denoising and quantitative analysis when compared with traditional DnCNN and Gaussian filters.
The overall organization and presentation of the manuscript are excellent. The figures (e.g., Figures 1–4) are clear and intuitive, and the supplementary video (Movie S1) effectively illustrates the methodology and results.
Although the manuscript is generally well-written, careful proofreading is still warranted for several details. Notably, the quality of the supplementary materials appears to be slightly lower than that of the main text, with minor issues in formatting and wording. The following detailed suggestions should be addressed prior to acceptance:
1. Computational Considerations (not currently addressed in the manuscript)
While the manuscript thoroughly demonstrates the efficacy of the proposed algorithm in denoising and enhancing image quality, a critical consideration for clinical applicability is the computational cost associated with image reconstruction or processing. It is strongly recommended that the authors provide information on processing times (e.g., for training, inference, or reconstruction of a single image or dataset), or at least discuss the potential clinical impact in the discussion section.
2. Training Dataset Size
As the authors appropriately acknowledge in the “Limitations” section, the training dataset includes only 29 patients. The reviewers commend the authors for explicitly recognizing this limitation, which reflects a rigorous and transparent research approach.
3. Supplementary Table S1: Inconsistent case count formatting
-
Original: (h) thoracic outlet syndrome (5), (i) cardiac disease (2), (k) inflammatory (2)
-
Issue: These entries lack “n=” notation, unlike others such as (a) suspected pulmonary embolism (n = 15)
-
Suggestion: Standardize formatting to (n = 5), (n = 2), etc.
4. Supplementary Text 1: Redundant wording
-
Original: Each branch applies 9 Dilated Convolution Block blocks.
-
Issue: Redundant repetition of “Block blocks.”
-
Suggestion: Revise to “9 Dilated Convolution Blocks” or “9 blocks of Dilated Convolution.”
5. Supplementary Text 2: Incomplete phrase
-
Original: where ℒ(𝒙, 𝒛, 𝒖) is the augmented Lagrangian for , 𝒚 is the original noisy image...
-
Issue: The phrase “for ,” is incomplete.
-
Suggestion: Revise to “for the optimization problem,” or another appropriate phrase.
6. Abstract: Countable noun usage
-
Original: Dataset of patients with suspected pulmonary diseases was used.
-
Issue: “Dataset” is a countable noun and requires an article or plural form.
-
Suggestion: Change to “A dataset of patients... was used” or “Datasets of patients... were used.”
7. Abstract: Terminology
-
Original: for balancing the noisy control and image fidelity.
-
Issue: “noisy control” is uncommon; should be “noise control.”
-
Suggestion: Use “for balancing the noise control and image fidelity.”
8. Abstract: Subject-verb agreement
-
Original: The fractal dimension and k-means segmentation of the pulmonary perfusion images was calculated...
-
Issue: Compound subject requires plural verb.
-
Suggestion: Change to “were calculated.”
9. Abstract: Redundancy
-
Original: ...overall image quality comparing as scored by two radiologists.
-
Issue: Redundant phrasing.
-
Suggestion: Use “...overall image quality as scored by two radiologists.”
10. Abstract: Ambiguous clause
-
Original: ...which showed the improvement of quantitative fractal analysis.
-
Issue: The reference of “which” is ambiguous.
-
Suggestion: Rephrase to “which led to improved quantitative fractal analysis” or “and demonstrated improved quantitative fractal analysis.”
11. Introduction: Subject-verb agreement
-
Original: one of the challenges of pulmonary MRI are low proton density...
-
Issue: Subject is singular (“one”), verb should be singular.
-
Suggestion: Use “is low proton density...”
12. Introduction: Incorrect word form
-
Original: To analyze the perfusion images, denoise is an important pre-process step.
-
Issue: Should use nouns “denoising” and “pre-processing.”
-
Suggestion: “...denoising is an important pre-processing step.”
13. Introduction: Formality and precision
-
Original: The noise in pulmonary perfusion MRI was especially spatially dependent...
-
Suggestion: Use “was particularly spatially dependent” or “was highly spatially dependent.”
14. Methods: Hyphenation
-
Original: ...during post processing.
-
Suggestion: Change to “post-processing.”
15. Inconsistency in input dimensionality (Figure 1A vs. text)
-
Issue: Figure 1A states that “3D noisy images” are input, while the text specifies that 2D slices are denoised due to GPU constraints.
-
Suggestion: Clarify whether inputs are 2D or 3D. If the model processes 2D slices, revise Figure 1A caption to “Each 2D noisy slice serves as an input to the PnP-BSN model.”
16. Model Design: Redundancy
-
Original: ...for enforcing the image fidelity.
-
Issue: Phrase is unnecessarily repeated.
-
Suggestion: Simplify to: “Here, the PnP-BSN method denoises the image by using AP-BSN as an image prior (in the second term of Eq. 1) and enforces image fidelity using the original noisy image (in the first term of the iterative PnP algorithm).”
17. Results: Typographical error
-
Original: ...as the vassal edges and complexity are better illustrated.
-
Issue: “vassal” is a misuse; correct term is “vessel.”
-
Suggestion: Use “vessel edges.”
18. Results: Typographical error
-
Original: ...over image quality...
-
Issue: Should be “overall image quality.”
-
Suggestion: Correct the typo.
19. Figure legend mislabeling
-
Original: ...overall image quality for MR tagging images...
-
Issue: Study focuses on pulmonary perfusion MRI, not MR tagging.
-
Suggestion: Revise to “pulmonary perfusion images” or “MR perfusion images.”
20. Discussion: Unnecessary comma
-
Original: ...a Gaussian denoising algorithm, for pulmonary contrast perfusion MRI.
-
Suggestion: Remove the comma.
21. Discussion: Incorrect noun form
-
Original: Future works should extend...
-
Suggestion: Use “Future work should extend...” or “Future studies should extend...”
22. Conclusion: Grammar
-
Original: Further this self-supervised learning framework...
-
Suggestion: Use “Furthermore, this self-supervised learning framework...” or “Additionally, this self-supervised learning framework...”
The manuscript is generally well-written and scientifically clear. However, several grammatical issues, inconsistent terminology, and minor typographical errors—particularly in the abstract and supplementary materials—should be addressed to improve overall readability and precision.
Author Response
We thank the reviewer for the time, effort, and attention given to our manuscript. The comments and suggestions were thoughtful and are addressed point-by-point below. We feel that addressing the reviewer’s concerns has significantly strengthened and improved the manuscript.
Reviewer #2 (Comments for the Author):
This study proposes a self-supervised denoising model based on a Plug-and-Play (PnP) framework incorporating the Asymmetric Pixel-shuffle Downsampling Blind-Spot Network (AP-BSN), aiming to enhance the image quality of pulmonary perfusion MRI. The manuscript effectively addresses the challenges of low signal-to-noise ratio (SNR) in dynamic contrast-enhanced (DCE) pulmonary MRI due to low proton density and short T2*, as well as the fundamental limitation of lacking “clean” images for supervised training. The model performs denoising without the need for clean image references by balancing noise control and image fidelity. Through radiologist-based scoring of SNR, sharpness, and overall image quality, as well as quantitative assessments using fractal dimension analysis and k-means clustering, the PnP-BSN approach demonstrates superior performance in both denoising and quantitative analysis when compared with traditional DnCNN and Gaussian filters.
The overall organization and presentation of the manuscript are excellent. The figures (e.g., Figures 1–4) are clear and intuitive, and the supplementary video (Movie S1) effectively illustrates the methodology and results.
Although the manuscript is generally well-written, careful proofreading is still warranted for several details. Notably, the quality of the supplementary materials appears to be slightly lower than that of the main text, with minor issues in formatting and wording. The following detailed suggestions should be addressed prior to acceptance:
R2C1. Computational Considerations (not currently addressed in the manuscript)
While the manuscript thoroughly demonstrates the efficacy of the proposed algorithm in denoising and enhancing image quality, a critical consideration for clinical applicability is the computational cost associated with image reconstruction or processing. It is strongly recommended that the authors provide information on processing times (e.g., for training, inference, or reconstruction of a single image or dataset), or at least discuss the potential clinical impact in the discussion section.
Response: We appreciate the reviewer’s insightful comment. We added the following information into the Data processing and evaluation section in Methods: “The model was trained for 20 epochs (approximately 23 hours), and the inference time per image was ~8 seconds.”
R2C2. Training Dataset Size
As the authors appropriately acknowledge in the “Limitations” section, the training dataset includes only 29 patients. The reviewers commend the authors for explicitly recognizing this limitation, which reflects a rigorous and transparent research approach.
Response: We appreciate the reviewer’s recognition of our transparent acknowledgment of the dataset size as a limitation. To more explicitly indicate this constraint, we have revised the text in the manuscript to state: “A limitation of the current study is the small number of exams in the training (n=29) and testing datasets (n=8),” which reflects our intention to clearly and rigorously disclose this limitation.
We would like to clarify that, although the number of unique subjects is limited, each dataset comprises full 3D lung image volumes. From the training datasets, a total of 3448 2D slices were extracted, and from the testing datasets, 1124 2D slices were derived. While the number of patients remains modest, the large number of spatially diverse slices provides substantial variability, which helps improve the robustness and generalizability of the deep learning model. We added this information in Methods Section as: “The 37 patient datasets were randomly divided 80:20 into training and independent testing datasets, with 29 subjects (3448 2D slices) for training and 8 subjects (1124 2D slices) for testing.”
R2C3. Supplementary Table S1: Inconsistent case count formatting
- Original: (h) thoracic outlet syndrome (5), (i) cardiac disease (2), (k) inflammatory (2)
- Issue: These entries lack “n=” notation, unlike others such as (a) suspected pulmonary embolism (n = 15)
- Suggestion: Standardize formatting to (n = 5), (n = 2), etc.
Response: Thank you for this comment. We have modified all the case count formatting to the consistent format of “n = …” as the following: “Eligible patient groups included those with (a) suspected pulmonary embolism (n = 15), (b) hypertension (n = 1), (c) chest pain (n = 6), (d) Marfan syndrome (n = 1), (e) dysphagia (n = 1), (f) Horner’s syndrome (n = 1), (g) Anisocoria (n = 1), (h) thoracic outlet syndrome (n = 5), (i) cardiac disease (n = 2), (j) superior vena cava (SVC) syndrome (n = 2) and (k) inflammatory (n = 2). Within the 37 patients, 5 of them were children (8 ± 8 years).”
R2C4. Supplementary Text 1: Redundant wording
- Original: Each branch applies 9 Dilated Convolution Block blocks.
- Issue: Redundant repetition of “Block blocks.”
- Suggestion: Revise to “9 Dilated Convolution Blocks” or “9 blocks of Dilated Convolution.”
Response: Thank you for this comment. We have revised it to “9 Dilated Convolution Blocks”.
R2C5. Supplementary Text 2: Incomplete phrase
- Original: where ℒ(?, ?, ?) is the augmented Lagrangian for , ? is the original noisy image...
- Issue: The phrase “for ,” is incomplete.
- Suggestion: Revise to “for the optimization problem,” or another appropriate phrase.
Response: Thank you for the comment. We have revised it to “where ℒ(?, ?, ?) is an augmented Lagrangian for the optimization problem, …”.
R2C6. Abstract: Countable noun usage
- Original: Dataset of patients with suspected pulmonary diseases was used.
- Issue: “Dataset” is a countable noun and requires an article or plural form.
- Suggestion: Change to “A dataset of patients... was used” or “Datasets of patients... were used.”
Response: Thank you for the comment. We have revised it to “A dataset of patients with suspected pulmonary diseases were used.”
R2C7. Abstract: Terminology
- Original: for balancing the noisy control and image fidelity.
- Issue: “noisy control” is uncommon; should be “noise control.”
- Suggestion: Use “for balancing the noise control and image fidelity.”
Response: Thank you for the comment. We have revised it to “for balancing the noise control and image fidelity.”
R2C8. Abstract: Subject-verb agreement
- Original: The fractal dimension and k-means segmentation of the pulmonary perfusion images was calculated...
- Issue: Compound subject requires plural verb.
- Suggestion: Change to “were calculated.”
Response: Thank you for the comment. We have revised it to “The fractal dimension and k-means segmentation of the pulmonary perfusion images were calculated for comparing denoising performance.”
R2C9. Abstract: Redundancy
- Original: ...overall image quality comparing as scored by two radiologists.
- Issue: Redundant phrasing.
- Suggestion: Use “...overall image quality as scored by two radiologists.”
Response: Thank you for the comment. We have revised it to “…, and overall image quality as scored by two radiologists.”
R2C10. Abstract: Ambiguous clause
- Original: ...which showed the improvement of quantitative fractal analysis.
- Issue: The reference of “which” is ambiguous.
- Suggestion: Rephrase to “which led to improved quantitative fractal analysis” or “and demonstrated improved quantitative fractal analysis.”
Response: Thank you for the comment. We have revised it to “…, which led to improved quantitative fractal analysis.”
R2C11. Introduction: Subject-verb agreement
- Original: one of the challenges of pulmonary MRI are low proton density...
- Issue: Subject is singular (“one”), verb should be singular.
- Suggestion: Use “is low proton density...”
Response: Thank you for the comment. We have revised it to “However, one of the challenges of pulmonary MRI is low proton density …”
R2C12. Introduction: Incorrect word form
- Original: To analyze the perfusion images, denoise is an important pre-process step.
- Issue: Should use nouns “denoising” and “pre-processing.”
- Suggestion: “...denoising is an important pre-processing step.”
Response: Thank you for the comment. We have revised it to “...denoising is an important pre-processing step.”
R2C13. Introduction: Formality and precision
- Original: The noise in pulmonary perfusion MRI was especially spatially dependent...
- Suggestion: Use “was particularly spatially dependent” or “was highly spatially dependent.”
Response: Thank you for the comment. We have revised it to “was particularly spatially dependent”.
R2C14. Methods: Hyphenation
- Original: ...during post processing.
- Suggestion: Change to “post-processing.”
Response: Thank you for the comment. We have revised it to “...during post-processing”.
R2C15. Inconsistency in input dimensionality (Figure 1A vs. text)
- Issue: Figure 1A states that “3D noisy images” are input, while the text specifies that 2D slices are denoised due to GPU constraints.
- Suggestion: Clarify whether inputs are 2D or 3D. If the model processes 2D slices, revise Figure 1A caption to “Each 2D noisy slice serves as an input to the PnP-BSN model.”
Response: Thank you for the comment. We have revised it to “Each 2D noisy slice serves as an input to the PnP-BSN model.”
R2C16. Model Design: Redundancy
- Original: ...for enforcing the image fidelity.
- Issue: Phrase is unnecessarily repeated.
- Suggestion: Simplify to: “Here, the PnP-BSN method denoises the image by using AP-BSN as an image prior (in the second term of Eq. 1) and enforces image fidelity using the original noisy image (in the first term of the iterative PnP algorithm).”
Response: Thank you for this comment. We have revised it to “Here, the PnP-BSN method denoises the image by using AP-BSN as an image prior (in the second term of Eq. 1) and enforces image fidelity using the original noisy image (in the first term of the iterative PnP algorithm).”
R2C17. Results: Typographical error
- Original: ...as the vassal edges and complexity are better illustrated.
- Issue: “vassal” is a misuse; correct term is “vessel.”
- Suggestion: Use “vessel edges.”
Response: Thank you for this comment. We have corrected the typo and revised it to “vessel edges”.
R2C18. Results: Typographical error
- Original: ...over image quality...
- Issue: Should be “overall image quality.”
- Suggestion: Correct the typo.
Response: Thank you for this comment. We have corrected the typo and revised it to “overall image quality”.
R2C19. Figure legend mislabeling
- Original: ...overall image quality for MR tagging images...
- Issue: Study focuses on pulmonary perfusion MRI, not MR tagging.
- Suggestion: Revise to “pulmonary perfusion images” or “MR perfusion images.”
Response: Thank you for this comment. We have revised it to “…, and overall image quality for pulmonary perfusion images…”.
R2C20. Discussion: Unnecessary comma
- Original: ...a Gaussian denoising algorithm, for pulmonary contrast perfusion MRI.
- Suggestion: Remove the comma.
Response: Thank you for this comment. We have removed the comma and revised it to “Prior studies have used a Gaussian denoising algorithm for pulmonary contrast perfusion MRI.”
R2C21. Discussion: Incorrect noun form
- Original: Future works should extend...
- Suggestion: Use “Future work should extend...” or “Future studies should extend...”
Response: Thank you for this comment. We have revised it to “Future work should extend…”.
R2C22. Conclusion: Grammar
- Original: Further this self-supervised learning framework...
- Suggestion: Use “Furthermore, this self-supervised learning framework...” or “Additionally, this self-supervised learning framework...”
Response: Thank you for this comment. We have revised it to “Additionally, this self-supervised learning framework...”.
Response to Comments on the Quality of English Language
The manuscript is generally well-written and scientifically clear. However, several grammatical issues, inconsistent terminology, and minor typographical errors—particularly in the abstract and supplementary materials—should be addressed to improve overall readability and precision.
Response: We appreciate the detailed feedback on grammar and terminology from Reviewer 2. The manuscript and supplementary materials have been revised accordingly. We also conducted a thorough review of the overall readability and made additional revisions where necessary.

Reviewer 3 Report
Comments and Suggestions for Authors
This is an interesting and potentially publishable manuscript (MS); prior to acceptance several points should be improved, please see the list below. Towards the end of the introduction the authors should concisely but clearly state what they do, how they do it (methods, approaches, models, etc.), and what exactly do they get as the results. A sectioned plan of the entire paper is to be presented here. The definition of all scoring factors and parameters quantified in the current study should be provided in the revised MS in their explicit analytical form.
Prior to denoising and to manipulation of the MRI images, i missed the detailed procedure how the images were taken and how the contrast and different color schemes were established. What kind of features are to be enhanced with these choices of colors? I also question myself whether the procedure will work also for other MRI images, for instance those from the brain tissues? This is important to know because the fractal dimensions and tortuosity of the brain-MRI images as well as the underlying mathematical models of diffusion behind those images can be universal for MRI images of tissues with a certain degree of dead-end labyrinth-like environments. Regarding the interpretation and modelling of the brain-MRI images the authors are encouraged to have a look and to mention in the revised MS the recent reference [ https://doi.org/10.1088/1361-6463/abdff0 ].
The biblio of the MS should certainly be extended, to emphasize both the classical pioneering studies in the field as well as the most recent related advances and model-based predictions.
Author Response
We thank the reviewer for the time, effort, and attention given to our manuscript. The comments and suggestions were thoughtful and are addressed point-by-point below. We feel that addressing the reviewer’s concerns has significantly strengthened and improved the manuscript.
Reviewer #3 (Comments for the Author):
This is an interesting and potentially publishable manuscript (MS); prior to acceptance several points should be improved, please see the list below.
R3C1. Towards the end of the introduction the authors should concisely but clearly state what they do, how they do it (methods, approaches, models, etc.), and what exactly do they get as the results. A sectioned plan of the entire paper is to be presented here.
Response: We thank the reviewer for this suggestion. In response, we have revised the final paragraph of the Introduction to clearly state the objective of the study, the methods used and the main findings.
The last paragraph in the Introduction is now rephrased as: “We sought to develop and evaluate a self-supervised deep learning framework for denoising pulmonary DCE MRI. Specifically, we integrate an AP-BSN within a Plug-and-Play (PnP) framework [12] incorporating the original noisy image to enhance image fidelity which a critical aspect for medical imaging applications and without re-quiring clean reference data. The model was trained on prospectively acquired 3D lung MRI datasets using 2D slices as inputs and evaluated through blinded radiologist scoring and fractal analysis of perfusion complexity which enables pixel-by-pixel estimation of perfusion image complexity while mitigating potential bias introduced by noise. In this technical proof-of-concept study, we show that PnP-BSN could improve image SNR without significant loss of sharpness and enhance pulmonary perfusion analysis.”
R3C2. The definition of all scoring factors and parameters quantified in the current study should be provided in the revised MS in their explicit analytical form.
Response: We thank the reviewer for this suggestion. In the revised manuscript, we have provided clear and explicit definitions for all scoring factors used in the radiologist-based evaluation. These details have been provided as Supplementary Information Text 4.
Three image quality criteria were assessed for each denoising method: signal-to-noise ratio (SNR), sharpness, and overall image quality. These were evaluated subjectively by two radiologists using a standardized 5-point Likert scale (1 being the worst and 5 being the best).
• SNR was defined as the radiologist’s perception of noise in the image, where a higher score indicates lower visible noise.
• Sharpness was defined as the perceived clarity of anatomical structures and the degree to which structural blurring was suppressed by the denoising algorithm.
• Overall image quality reflected a combined impression of noise, structural fidelity, and sharpness, using the original noisy image as a reference.
Scoring was performed using the following 5-point scale:
(1) Poor: high noise and/or significant blurring;
(2) Fair: reduced noise and/or significant blurring;
(3) Good: low noise but moderate blurring;
(4) Very good: low noise and well-controlled blurring;
(5) Excellent: no significant noise or blurring.
Radiologists viewed one PowerPoint slide per patient, which displayed all the slices of denoised images from the same subject but from four different methods (Methods 1–4) presented in random order from left to right. The original noisy image was also shown as a reference. Each method was scored independently in all three metrics, and the scores were recorded using a structured Excel sheet for later statistical analysis.
R3C3. Prior to denoising and to manipulation of the MRI images, i missed the detailed procedure how the images were taken and how the contrast and different color schemes were established. What kind of features are to be enhanced with these choices of colors?
Response: We thank the reviewer for this important observation. We have revised the Methods section of the manuscript to provide additional details regarding both image acquisition and the visualization process used for qualitative evaluation.
a. Image acquisition:
Dynamic contrast-enhanced (DCE) lung MRI was performed using 1.5 T or 3.0 T scanners with a time-resolved 3D spoiled gradient-echo (GRE) sequence. A gadolinium-based contrast agent (0.1 mmol/kg) was administered intravenously, and image acquisition was initiated shortly after injection, timed to capture the peak pulmonary perfusion phase during a patient breath-hold.
We have clarified this in Methods section as: “In total, 5 datasets were acquired using 3.0 T scanners (MAGNETOM VIDA, Siemens Healthcare, Erlangen, Germany), and 32 were acquired using 1.5 T scanners (MAG-NETOM Aera, Siemens Healthcare, Erlangen, Germany). For each individual, pulmonary perfusion images were acquired with a time-resolved 3D spoiled gradient-echo sequence. A gadolinium-based contrast agent (0.1 mmol/kg) was administered intravenously, and image acquisition was initiated shortly after injection, timed to capture the peak pulmonary perfusion phase during breathholds.”
b. Original images and denoised images:
“All image denoising and evaluation were performed on grayscale images. For visualization, a consistent linear grayscale window was applied to preserve relative intensity differences and enable visual comparison.”
We added this in Supplementary Information Text 4: Full details of radiologist scoring guidelines.
In Figure 3 caption, we added “Red, green, and purple correspond to high-, medium-, and low-intensity regions, respectively.” The colors are for indicating the different segmention regions.
c. Fractal analysis visualization:
For visualizations, pseudo-color maps (e.g., ‘hot’ or ‘jet’) were applied solely for illustrative purposes. The chosen contrast settings and color maps were selected to emphasize regions of contrast enhancement, vascular structure, and perfusion heterogeneity in the lung region. These settings were fixed across all methods and cases to ensure unbiased visual comparison of image quality and noise suppression.
We added the above information in Supplementary Information Text 3. Full details of fractal analysis and fractal dimension.
R3C4. I also question myself whether the procedure will work also for other MRI images, for instance those from the brain tissues? This is important to know because the fractal dimensions and tortuosity of the brain-MRI images as well as the underlying mathematical models of diffusion behind those images can be universal for MRI images of tissues with a certain degree of dead-end labyrinth-like environments. Regarding the interpretation and modelling of the brain-MRI images the authors are encouraged to have a look and to mention in the revised MS the recent reference [https://doi.org/10.1088/1361-6463/abdff0].
Response: We thank the reviewer for this thoughtful and forward-looking comment. We agree that the proposed framework, particularly the use of fractal analysis to quantify perfusion heterogeneity, may have broader applicability to other tissue types imaged by MRI. For example, in our manuscript, we have referenced that the myocardial perfusion1 and CT pulmonary images2 are the potential applications of fractal analysis.
1. Tanabe N, Sato S, Suki B, Hirai T. Fractal Analysis of Lung Structure in Chronic Obstructive Pulmonary Disease. Front Physiol. 2020;11:603197.
2. Michallek F, Dewey M. Fractal analysis of the ischemic transition region in chronic ischemic heart disease using magnetic resonance imaging. Eur Radiol. 2017;27(4):1537–46.
We have reviewed the suggested reference ([https://doi.org/10.1088/1361-6463/abdff0]) and found fractal analysis works in brain. We have added a brief discussion and citation of the suggested reference in the revised Discussion section to acknowledge this connection and its potential implications.
“While our study focuses on pulmonary perfusion, the use of fractal analysis to quantify image complexity may be applicable to other tissues, such as the brain1,2.”
1. Esteban FJ, Sepulcre J, de Mendizabal NV, et al. Fractal dimension and white matter changes in multiple sclerosis. Neuroimage. Jul 1 2007;36(3):543-9. doi:10.1016/j.neuroimage.2007.03.057
2. Reishofer G, Studencnik F, Koschutnig K, Deutschmann H, Ahammer H, Wood G. Age is reflected in the Fractal Dimensionality of MRI Diffusion Based Tractography. Sci Rep. Apr 3 2018;8(1):5431. doi:10.1038/s41598-018-23769-6
R3C5. The biblio of the MS should certainly be extended, to emphasize both the classical pioneering studies in the field as well as the most recent related advances and model-based predictions.
Response: We thank the reviewer for this valuable suggestion. In response, we have revised and expanded the bibliography to include both classical image denoising works and recent advances in MRI denoising, particularly in brain imaging, where most of the relevant literature is concentrated. To the best of our knowledge, only limited work has been published specifically on pulmonary MRI denoising. For example, we now cite Dabov et al. (2007)3, who introduced the classical BM3D framework that remains a reference standard in traditional denoising, and Ulyanov et al. (2020)4, who proposed the Deep Image Prior (DIP) model, a foundational unsupervised deep learning approach that demonstrated the effectiveness of network structure itself as an image prior. A recent review of deep learning denoising methods in medical images5.
“Conventional methods such as BM3D remain widely used in medical image denoising 3 due to their robustness and interpretability, while unsupervised denoising networks like Deep Image Prior 4 have demonstrated the potential of leveraging network structure as an image prior. More recently, a range of deep learning-based methods have been proposed for denoising in medical imaging5; however, relatively few studies have focused specifically on pulmonary MRI, highlighting the need for and significance of our proposed approach.”
1. Esteban FJ, Sepulcre J, de Mendizabal NV, et al. Fractal dimension and white matter changes in multiple sclerosis. Neuroimage. Jul 1 2007;36(3):543-9. doi:10.1016/j.neuroimage.2007.03.057
2. Reishofer G, Studencnik F, Koschutnig K, Deutschmann H, Ahammer H, Wood G. Age is reflected in the Fractal Dimensionality of MRI Diffusion Based Tractography. Sci Rep. Apr 3 2018;8(1):5431. doi:10.1038/s41598-018-23769-6
3. Dabov K, Foi A, Katkovnik V, Egiazarian K. Image denoising by sparse 3-D transform-domain collaborative filtering. IEEE Trans Image Process. Aug 2007;16(8):2080-95. doi:10.1109/tip.2007.901238
4. Ulyanov D, Vedaldi A, Lempitsky V. Deep Image Prior. Int J Comput Vision. Jul 2020;128(7):1867-1888. doi:10.1007/s11263-020-01303-4
5. Nazir N, Sarwar A, Saini BS. Recent developments in denoising medical images using deep learning: An overview of models, techniques, and challenges. Micron. May 2024;180:103615. doi:10.1016/j.micron.2024.103615

Round 2
Reviewer 1 Report
Comments and Suggestions for Authors
Thank you for revising the manuscript. I have further comments for the clarity of the some issues.
- Please correct the phrase "37 datasets" at line 84.
- You may not compare the acquisition parameter's difference using T-Test. Please remove or revise the statements starting at lin 91.
- Please provide details for the training of the model. Did you use transfer learning? How did you optimize the hyperparameters?
Author Response
Reviewer #1 (Comments for the Author):
R1C1. Please correct the phrase "37 datasets" at line 84.
Response: We thank the reviewer for noting this wording issue. We have revised 37 datasets to 37 subjects in line 84.
R1C2. You may not compare the acquisition parameter's difference using T-Test. Please remove or revise the statements starting at lin 91.
Response: We have removed the sentence starting at line 91 as suggested.
R1C3. Please provide details for the training of the model. Did you use transfer learning? How did you optimize the hyperparameters?
Response: We appreciate the reviewer’s interest in the details of our model training. We did not use transfer learning in this study. Instead, we trained the network from scratch, following the training strategy and hyperparameter settings described in the original AP-BSN paper.

Reviewer 2 Report
Comments and Suggestions for Authors
The author has provided a thorough explanation and made detailed revisions to the manuscript. At this point, I have no further questions, and I will leave the subsequent decisions to the editor's discretion.
Author Response
We sincerely thank the reviewer for the valuable feedback and constructive comments, which have helped improve the quality of our manuscript.
Reviewer 3 Report
Comments and Suggestions for Authors
There were indeed some improvements of the text; the current revision however is still not on the highest level, in particular regarding the comparison of the novelty of the presented results to the state of the art in the literature (the discussion section). Also, the original relevant reference in JPhyD from 2021 regarding the theoretical models of MRI and time-dependent diffusion is not cited. The authors should try to improve the text and the bibliography further and then upload the second revision.
Author Response
R3C1. There were indeed some improvements of the text; the current revision however is still not on the highest level, in particular regarding the comparison of the novelty of the presented results to the state of the art in the literature (the discussion section). Also, the original relevant reference in JPhyD from 2021 regarding the theoretical models of MRI and time-dependent diffusion is not cited. The authors should try to improve the text and the bibliography further and then upload the second revision.
Response:
We thank the reviewer for this insightful comment. We have modified and added the following paragraph to further clarify how our presented results compare to state-of-the-art methods:
“More recently, a range of deep learning-based methods have been proposed for denoising in medical imaging; however, relatively few studies have focused specifically on pulmonary MRI when paired clean images are unavailable, highlighting the need for a self-supervised learning approach. Our work demonstrates the feasibility of self-supervised denoising for pulmonary perfusion MRI. Unlike methods such as Gaussian filtering, BM3D, or supervised deep learning approaches that require paired high-quality training data, and unlike the blind-spot network that assume pixel-independent noise, our approach can denoise without clean reference images and adapt to the spatially varying noise characteristics of pulmonary imaging. We address the challenge of spatially varying noise by employing AP-BSN within a PnP framework to improve image fidelity. The subsequent fractal analysis of the denoised images provides a quantitative assessment, revealing the fractal dimension in pulmonary perfusion MRI and suggesting that denoising enhances quantitative pulmonary image analysis.”
JPhyD from 2021 has been cited.
